

# Cryostratigraphy, sedimentology and the late Quaternary evolution of the Zackenberg Delta, Northeast Greenland

Graham L. Gilbert[1,2], Stefanie Cable[3], Christine Thiel[4,5], Hanne H. Christiansen[1], Bo Elberling[3]

[1]Arctic Geology Department, University Centre in Svalbard, P.O. Box 156, 9170, Longyearbyen, Norway
[2]Department of Earth Science, University of Bergen, Realfagbygget, Allegt. 41, 5007, Bergen, Norway
[3]Center for Permafrost (CENPERM), Department of Geosciences and Natural Resource Management, University of Copenhagen, Oester Voldgade 10, 1350, Copenhagen, Denmark
[4]Nordic Laboratory for Luminescence Dating, Department of Earth Sciences, Aarhus University, Risø DTU, DK-4000 Roskilde, Denmark.
[5]Leibniz-Institut for Applied Geophysics, University of Tübingen, Geschwister-Scholl-Platz, 72074, Tübingen, Germany

*Correspondence to*: Graham L. Gilbert (graham.gilbert@unis.no)

Keywords: Permafrost, Cryofacies, Valley fill, Fjord delta, Greenland

## Abstract

The Zackenberg Delta is located in Northeast Greenland (74°30'N, 20°30'E) at the outlet of the Zackenberg Valley. The
deltaic fill at the mouth of the valley consists of a series of terraces (ca. 2 km[2]) formed during a fall in relative sea level. The modern Zackenberg River has incised through the paleo-deltaic deposits creating exposures (up to 22 m in height) along the rivers banks. In addition, coastal processes have exposed sediments in 4 m – 20 m high coastal cliffs. In 2012, two 20 m long ice-bonded sediment cores were retrieved from within the deltaic deposits. The combination of river and coastal exposures with the analysis of ground ice in these cores permitted the reconstruction of the valley-fill succession and evaluation of the
timing and nature of permafrost aggradation. Permafrost in the palaeo-deltaic deposits is predominantly epigenetic and aggraded following the subaerial exposure of the delta plain (beginning ca. 11 ka). The exposed deposits in the Zackenberg Valley provide a unique opportunity to investigate the relationship between depositional environments and processes, grain-size properties and cryostratigraphy in epigenetic lowland permafrost environments.



## 1 Introduction

Formerly glaciated valleys and fjords are sedimentary depocenters in which large volumes of sediment have accumulated during the late Weichselian and early Holocene (Aarseth, 1997; Hansen, 2001; Eilertsen et al., 2011). Fjord valley-fills predominantly develop during highstand and relative sea-level fall following deglaciation, when sediment yield is high and

accommodation space is declining (Ballantyne, 2002; Corner, 2006). The formerly ice-covered areas of coastal Greenland experienced isostatic rebound following deglaciation (Fleming and Lambeck, 2004). This has resulted in uplifting, incision and erosion of valley-fill deposits by Holocene fluvial and coastal activity. Recent studies have examined the deltaic infill of fjords in high-relief landscapes (Hansen, 2004; Corner, 2006; Eilertsen et al., 2011; Marchand et al., 2013). However, few of these studies were in landscapes with permafrost. Therefore, the relationship between ground ice and the depositional

setting, in high-relief landscapes has received less attention. Combining sedimentological observations with the systematic classification of ground ice provides a mechanism to correlate sedimentary facies with cryofacies – resulting in an improved palaeoenvironmental reconstruction.

    Cryostratigraphy is the description, interpretation and correlation of cryofacies and their relationship to the host deposits (French and Shur, 2010; Murton, 2013). Systematic classification of cryofacies permits the differentiation between

syngenetic and epigenetic permafrost (Gilbert et al., 2016). In syngenetic settings, permafrost aggrades upwards at a rate proportional to the sedimentation rate at the ground surface. Ground ice in syngenetic permafrost primarily forms as segregated ice at the top of permafrost (Mackay, 1972). Conversely, epigenetic permafrost aggrades downwards after the deposition of the host material. Where moisture and sediment conditions permit, syngenetic permafrost contains a diverse suite of ice-rich cryofacies. Conversely, epigenetic permafrost is characteristically ice-poor as the moisture source is

restricted to the surrounding sediment (French and Shur, 2010; Murton, 2013). The presence of the pore cryofacies in frost susceptible material is characteristic of epigenetic permafrost (Stephani et al., 2014). However, ice-rich cryofacies may form in epigenetic permafrost if an external water source is available to recharge the local groundwater system (Pollard, 2000b; Kanevskiy et al., 2014). In addition to the mode of permafrost aggradation, sediment characteristics and the availability of moisture have a controlling influence on the presence and morphology of ground ice (Stephani et al., 2014). The application

of cryostratigraphy to palaeo-landscape reconstruction therefore requires consideration of the physical properties of the soil, sediment, or bedrock which host ground ice.

    This study reconstructs Holocene permafrost and landscape change in the Zackenberg lowlands using sedimentary and cryofacies. The objectives are: (1) to describe sedimentary facies and cryofacies as observed in sections and cores; (2) to relate ground ice formation in permafrost to sediment properties and depositional environments; and (3) to combine

observations to reconstruct landscape change in the Zackenberg lowlands since the Last Glacial Maximum. This is the first study to investigate cryostratigraphy in northeast Greenland, and sheds new light on the variability of ground ice in high-relief Arctic landscapes.





## 2 Regional setting

### 2.1 Glaciation, deglaciation and relative sea level change

Zackenberg is located on the Wollaston Foreland in Northeast Greenland (74°30'N, 20°30'E; Figure 1), 90 km east of the Greenland Ice Sheet margin. This region has been glaciated several times during the Quaternary period (Hjort, 1981; Funder

et al., 1994; Bennike et al., 2008). The last glaciation culminated in the Last Glacial Maximum (LGM) during the late Weichselian, when Greenland Ice Sheet extended across the region (Bennike et al., 2008). Sedimentary archives and the geomorphology of the continental shelf and slope indicate that during the LGM, ca. 22 ka, grounded glacier ice extended onto the outer continental shelf (Evans et al., 2002; Ó Cofaigh et al., 2004; Winkelmann et al., 2010). Streamlined subglacial bedforms record the presence of erosive, warm-based ice streams in the major fjords and cross-shelf troughs in NE

Greenland during this period. Intervening areas were covered by grounded, but non-streaming ice, in some cases preserving pre-existing landforms (Funder et al., 2011). Deglaciation initiated in the coastal areas east of Young Sound between 10.1 and 11.7 ka (Bennike et al., 2008). The inner-fjord areas were ice free by between 9.5 and 7.5 ka. At Zackenberg, the earliest postglacial radiocarbon date from a marine fossil is 10.1 ka, suggesting the study area was ice-free by this time (Bennike et al., 2008).

Following deglaciation, the sea inundated low-lying areas. Rapid emergence is documented during the Early Holocene due to postglacial crustal rebound. Regional relative sea-level (RSL) curves are reconstructed from the height of raised beach deposits using optically stimulated luminescence dating (OSL) and AMS $^{14}$C dating of palaeosurfaces and marine fossils (Bennike and Weidick, 2001; Pedersen et al., 2011). Bennike et al. (2008) reconstructed RSL at Zackenberg using a combination of geomorphological evidence and radiocarbon dates of marine fossils (sub littoral bivalves). This curve

indicates that a Late Quaternary marine limit of ca. 70 m a.s.l. was attained at 10.1 ka. During the early Holocene, RSL declined rapidly, reaching present sea level by ca. 5 – 6 ka (Christiansen et al., 2002).

The erosional competence of ice streams during the LGM, deglaciation history and subsequent changes in RSL have important implications for both landscape and permafrost development in Zackenberg. Erosive ice streams in the fjords removed much of the sedimentary record from previous glacial-interglacial cycles. At the same time, warm-based glaciers

precluded the formation or preservation of permafrost due to frictional heat generated at the base of the sliding glacier and trapping of geothermal heat under the ice (Humlum, 2005). The age of permafrost in the Zackenberg Lowlands is likely linked to the timing of regression as warm boundary conditions at the sea floor would have prevented permafrost aggradation.

### 2.2 Climate and permafrost

Mean annual air temperature at Zackenberg between 1996 and 2013 was -9.0 °C. The mean annual precipitation for the same period was 219 mm water equivalent, of which 90 % falls as snow or sleet. Sea ice covers the Young Sound between 9-10 months each year from late-October (Hansen et al., 2008; Jensen et al., 2014). Hydrology and sediment transport in the

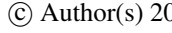



Zackenberg River have been summarized by Hasholt et al. (2008). The drainage basin covers an area of 512 km$^2$, of which 20% is glaciated. Runoff in the river typically begins in June and continues until September. Sediment transport is dominated by extreme discharge events. In addition to the Zackenberg River, a number of minor rivers and streams drain the hillslopes and lowland area.

Zackenberg is located within the continuous permafrost zone. The permafrost thickness is modeled to be between 200 m and 400 m (Christiansen et al., 2008). Ground temperatures are monitored to a depth of 20 m at two locations within the study area (C1 and C2 in Figure 1C). Temperatures at 20 m depth are ca. -6 °C and with little interannual variation since monitoring began in 2012. Seasonal thaw progression and active-layer thickness has been measured since 1996 (Christiansen et al., 2003). On average, the active-layer thickness varies between 70 cm and 80 cm – with topography, and its impact on

the snow regime and hydrology, being the primary controlling factor (Christiansen et al., 2008).

**2.3 Geomorphology and Quaternary geology**

The Zackenberg Valley is oriented along a NW trending fault (Koch and Haller, 1971; Nøhr-Hansen et al., 2011), separating Caledonian gneiss and granite exposed to the west in the Zackenberg Mountain (1303 m a.s.l.) from Cretaceous and Jurassic sedimentary rocks exposed to the east in the Aucellabjerg Mountain (911 m a.s.l.). The valley is ca. 8 km long, expanding

from 2 km to 7 km in width towards Young Sound (Figure 1C). The Zackenberg Lowland area is defined as the portion of the landscape below the Early Holocene marine limit (70 m a.s.l.). The lowlands are primarily a relict landscape, formed by glacial, fluvial, marine, and periglacial processes during glaciation and deglaciation and RSL decline. Glacial and glaciofluvial landforms include moraine ridges, ground moraine, meltwater plains, and raised delta terraces (Christiansen and Humlum, 1993; Christiansen et al., 2002). Permafrost landforms include ice-wedge polygons and palsas (Christiansen,

1998a; Christiansen et al., 2008).

The raised palaeo-delta at the mouth of the Zackenberg Valley covers an area of ca. 2 km$^2$ and consists of a series of terraces. To the west, the spatial extent of the paleo-delta is restricted by the Zackenberg Mountain. The eastern boundary is less easily defined and grades into glacial and glaciofluvial deposits. Christiansen et al. (2002) determined that the deltaic deposits began to form prior to 9.5 ka.

**3 Methods**

Data used for this study consists of sedimentological logs from nine river and coastal sections in the delta, geomorphological observations, and cryostratigraphic and sedimentological analysis of two 20 m long, ice-bonded sediment cores. Field data was collected over three summer field campaigns, in 2012, 2013 and 2015.

The natural exposures along the coast and river were examined to determine the sedimentary stratigraphy of the valley

deposits (Figure 1C; Figure 2). Sections, 4 m to 22 m high, were photographed and described in August 2015. Measurements included descriptions of sediment type, grain size, sorting, sedimentary structures and fossil content. It was not possible to



describe ground ice conditions in the sections as the rate of thaw perpendicular to the exposed surface outpaced the rate of backwasting due to erosion.

Two 20 m deep boreholes (C1 and C2) were drilled in September 2012 using a drill rig equipped with a 42 mm core barrel (Gilbert et al., 2015). The sites are located within Holocene delta terraces, ca. 500 m apart (Figure 1C). Core C1 is

located at 38 m a.s.l. At C1, ca. 12 m of undisturbed (frozen) core was retrieved across the 20 m interval. Disturbed (unfrozen) material was concentrated in the unfrozen active-layer and in the diamicton encountered below ca. 12 m depth. C2 (28 m a.s.l.) is situated at the base of a small slope, 3 m high. A seasonal snowbank accumulates at this site, which is located in the lee of the dominant northerly wind direction. Sand-rich surface deposits reflect nival-fluvial and aeolian sediment transport. At C2, 14 m of undisturbed core was recovered. Disturbed samples are spread across the length of the

core with missing intervals ranging up to 4 m in thickness. Retrieved cores were sealed in sterile plastic bags and stored in a freezer onsite. Frozen samples were transported to the University of Copenhagen for laboratory analysis.

In the laboratory, the C1 and C2 were split lengthwise, described and photographed. Cryostructures were classified using a system adapted from (Murton, 2013) and (French and Shur, 2010). Excess ice content and gravimetric ice content was calculated for ca. 180 samples. Samples were selected to account for vertical variations in ground ice and sediment

characteristics. Excess ice content ($E_i$; %), the water content in excess of the pore volume upon thawing, was estimated using the volumes of saturated sediment and supernatant water using (Kokelj and Burn 2005: Eq. 1):

$$E_i = \frac{(W_v \times 1.09)}{(S_v + (W_v \times 1.09))} \times 100$$

where $W_v$ is the volume of supernatant water, $S_v$ is the volume of saturated sediment and 1.09 is used to estimate the equivalent volume of ice. Gravimetric ice content ($G_i$; %), the ratio of the mass of water to the mass of dry sample, was estimated using the wet weight and dry weight of each sample using (Murton 2013: Eq. 1):

$$G_i = \frac{(M_i - M_d)}{M_d} \times 100$$

where $M_i$ is the mass of the frozen sample and $M_d$ is the mass of the sample following oven drying (90 °C; 24 hrs).

Optically Stimulated Luminescence (OSL) dating was performed on 14 samples from depths of up to 12 m on sediment from C1 and C2. Samples were only selected from intact, undisturbed cores. These ages are based on sand-sized quartz and feldspar grains, collected under subdued orange light conditions. Samples were processed at the Nordic Centre for Luminescence Dating at the University of Aarhus (Risø National Laboratory, Denmark).

**4 Description and interpretation of sedimentary facies and facies associations**

Nine sedimentary facies were identified (Table 1). These were defined based on the bulk macroscopic properties of the sediment – texture, structure, and bed boundaries. The facies ranged in lithology from a diamicton (facies 1) to silts, sands, and gravels (facies 2 to 9). Images of each facies are presented in Figure 3 and Figure 4. The facies were arranged into four





facies association (FA I – FA IV). Each facies association represents a distinct depositional environment. The facies associations are super-positioned in ascending order. FA I – the glacial facies association – occupies the lowermost position where present. The base of FA I was not observed. FA I is overlain by FA II and FA III. Together, FA II (the fjord-basin facies association) and FA III (the delta-slope facies association) form an upwards-coarsening succession. FA IV – the terrace top facies association – is the uppermost unit where present. The distribution of these facies associations is presented in Figure **5**.

### 4.1 FA I – Glacial facies association

FA I was observed at all sites except those bordering the present-day shoreline (S1 – S5). Exposures of FA I ranged up to 6 m in thickness and consisted of a compact, sandy, matrix-supported diamicton (facies 1). The unsorted deposits contained angular to subrounded clasts, up to boulder-size, of varying provenance. The deposits were over-consolidated when compared to other sedimentary facies. The lower boundary was not observed. The upper boundary to FA II was sharp and decreases in elevation, from ca. 41 m in the upper part of the delta to ca. 12 m towards the modern-day coastline (Figure **5**).

FA I is interpreted as a subglacial till on the basis of its stratigraphic position, compaction, lithology and lateral extent (Evans and Benn, 2004). The decline in elevation of FA I towards the southeast likely reflects the increasing depth to bedrock. Widespread till deposits have previously been identified in the Zackenberg valley bottom by Christiansen and Humlum (1993) and Christiansen et al. (2002) and outcrop at the ground surface in the glacial and glaciofluvial lowlands to the east of the delta terraces (Figure 1). FA I was likely deposited during the last glaciation when the Zackenberg lowlands were covered by an ice stream.

### 4.2 FA II – Fjord-basin facies association

FA II includes silts (facies 2), interbedded sands and silts (facies 3), and normally graded sands and silts (facies 4). Laminated sands (facies 6) and massive sands (facies 7) are recorded in the lower portion of FA II at S6 and S7. FA II outcrops at all locations with the exception of S3, S4, and S9 (Figure **5**). Individual beds appear horizontal. The thickness of this unit ranged between 2 and 10 m. The upper boundary is transitional when overlain by FA III and sharp and erosional when overlain by FA IV. At S6, C1 and C2, FA II initially fines upwards before an upwards coarsening into FA III (Figure **5**). At these sites, coarse-grained facies (facies 6 and 7) were observed at the base of FA II. In other sections, FA II generally fines upwards. Bioturbation was uncommon and restricted to facies 2. Macrofossils consisted of bivalve shells and shell fragments. Isolated outsized clasts were observed.

FA II is polygenetic and was deposited into a fjord environment during relative sea level highstand during and following deglaciation. The deposits consist of parallel-bedded fine sands and silts and reflect deposition by suspension fallout from hyperpycnal plumes and dilute turbidity currents (Table 1). The upwards fining and thinning observed at S6 and in C1 and C2 records the withdrawal of the sediment source as the ice front retreated north during deglaciation towards its grounding line at the mouth of the Zackenberg Valley (Ó Cofaigh et al., 1999). The upper portion of FA II is characterized by an




upwards coarsening and thickening. This unit is interpreted as the prodelta environment. The high-sand content of deposits from suspension is typical of steep, shallow-water deltas, where sand is carried beyond the delta slope by hypopycnal plumes (Corner, 2006). Turbidites record instabilities on the surrounding slopes, periods of high-river discharge, variations in glacier front position, or collapsing sediment plumes (Gilbert, 1983; Hansen, 2004).

The relative absence of bioturbation and limited number of trace-making organisms indicates a stressed environment with turbid water conditions and relatively high sedimentation rates (Netto et al., 2012). High-sedimentation rates are supported by the scattered observations of out-sized clasts as these would have been diluted by other sediment inputs. Overall, the vertical changes in the relative dominance of the facies records an approaching delta front.

**4.3 FA III – Delta-slope facies association**

FA III consists of interbedded sands and silts (facies 3), graded sands and silts (facies 4), laminated sands (facies 6), massive pebbly sands (facies 7), and stratified pebbly sands and gravels (facies 8). FA III is exposed at S2-S6 and in both C1 and C2. Exposures range in thickness from 2 to 10 m (Figure **5**). The relative influence of facies 3 and facies 4 decreases upwards as the unit coarsens and beds increase in thickness. Evidence for soft-sediment deformation is occasionally observed in association with dewatering structures. At sites S2-S6, beds dip 5° – 25° southeast, towards Young Sound (Figure 4a). Dip
angles increase upwards as the beds coarsen and thicken towards the top of the exposures.

The inclined, planar beds of FA III are interpreted as foresets deposited during the progradation of a Gilbert-type delta. The overall decline in the elevation towards Young Sound indicates that this facies association was deposited during relative sea level fall. Variations in gain size and facies composition reflect differences in their proximity to fluvial channels, variations in discharge, and sediment transport processes. The deposits reflect suspension settling, turbidity currents, debris
flows, and grain flows associated with slope failure on the delta front. Facies 7 and 8 were deposited by gravitational avalanches and debris flows following accumulation on the upper delta slope (Nemec, 1990; Plink-Björklund and Ronnert, 1999; Hansen, 2004). Facies 3, 4, and 6 record the activity of turbidity currents either due to underflow of sediment-laden river water or flow-transformation (Kneller and Buckee, 2000; Winsemann et al., 2007). The presence of soft-sediment deformation structures and absence of bioturbation suggests a rapid sedimentation rate for FA III.

**4.4 FA IV – Terrace-top facies association**

FA IV consists of cross-stratified sands (facies 5), laminated sands (facies 6), massive pebbly sands (facies 7), and gravels (facies 9). The facies association ranges between 0.5 and 3.0 m in thickness and is exposed at elevations between 5 and 45 m a.s.l. The elevation of FA IV declines towards Young Sound both to the south and east (Figure **5**). Facies 9 increases in dominance with distance inland. FA IV has a sharp, erosive contact with the underlying unit.
FA IV is polygenetic and the component facies are related to a range of depositional processes and environments. Deposits are interpreted to primarily reflect fluvial activity associated with a deltaic distributary plain or braided-river system. Diffusely stratified gravels with imbrication (facies 9) and planar parallel laminated sands (facies 6) indicate bedload





transport by unidirectional currents of varying strength (Miall, 2010). Cross-stratified sand beds record the migration of dunes in braided river channels or distributary systems (Miall, 2006; Winsemann et al., 2007). Massive pebbly sands (facies 7) are deposits of pseudoplastic debris flows (Miall, 2010). Secondary processes include winnowing by wind, marine reworking during relative sea-level fall, and redistribution of sediment by snowmelt (Christiansen and Humlum, 1993;

5 Christiansen, 1998b; Christiansen et al., 2002). FA IV developed at successively lower levels during relative sea-level fall and fluvial incision and channel migration.

## 5 Cryofacies

Three cryofacies are visually identified in C1 and C2 based on the bulk macroscopic characteristics of ground ice – namely the morphological expression of ice and the proportion of ice to sediment. These cryofacies are: (1) pore cryofacies, (2) layer

10 cryofacies, and (3) suspended cryofacies. An example of each cryofacies is given in Figure 6. In addition, disturbed sections were identified where samples were thawed during the drilling process. In these cases, ice present within the samples melted and the cryofacies were destroyed. The vertical distributions of the cryofacies and disturbed intervals are given in Figure 7 and Figure 8.

The pore cryofacies (Po) dominates at both C1 and C2. Po develops due to the *in situ* freezing of pore water in the spaces

15 between mineral grains, cementing individual grains. The interstitial ice is not visible to the unaided eye. The layer cryofacies (La) occurs various lithologies in both C1 and C2. Ice layers are less than 5 cm in thickness and may contain sediment grains or aggregates. La is observed in FA II and FA III in C1, and in FA III in C2. The suspended cryofacies (Su) develops where sediment grains or aggregates are suspended in ice. The visible ice content of Su exceeds 50%. Su is observed in FA II at C1. The variation between La and Su likely reflects differences in the freezing rate and availability of

20 water during permafrost aggradation (Calmels et al., 2012). It is hypothesized that both La and Su form from the injection of pressurized water in sediment during epigenetic permafrost formation (Murton, 2013).

Ground-ice content varies with cryofacies. Gravimetric moisture content and excess ice content were highest in intervals with either La or Su. Sections with Po were devoid of excess ice and were typified by low gravimetric moisture contents. At C1, the gravimetric moisture content of core samples with Po ranged from 20 to 48%, with no clear variation in depth.

25 Samples from intervals with La and Su ranged from 56 to 274% (Figure 7). Where present, excess ice content ranged from 2 to 52%. At C2, gravimetric moisture content in samples with Po were uniformly low and ranged between 11 and 48 % - no excess ice was observed in these samples (Figure 8). One La sample containing excess ice (5 %) was obtained from 3 m depth (24 m a.s.l.) in C2. Analysis of disturbed sections was not conducted as the natural moisture content was altered during drilling.



## 6 Geochronology

The OSL results of the 14 samples from C1 and C2 indicate a late-Weichselian to Holocene depositional age of the sediment in FA II, FA III, and FA IV (Table 2). Results from C1 suggest that the deposits formed between 13 and 12 ka (Figure 7). The lowermost six samples at C2 (below 26 m a.s.l.) indicate that the sediments were deposited between 12 and 10 ka. However, the two uppermost samples are considerably younger – suggesting that the site continued to aggrade during the Holocene (Figure 8).

These results indicate that the majority of the sedimentation in the valley-bottom took place in a narrow time interval following deglaciation. This is further substantiated by Christiansen et al. (2002), who presented [14]C AMS dating results for samples from FA II at S5. These results suggest that fjord-basin deposits were aggrading at S1 from approximately 9.5 ka. The OSL ages indicate that the Zackenberg lowlands were deglaciated approximately 1000 to 2000 years earlier than previous studies using radiocarbon dates from marine molluscs (Bennike et al., 2008).

## 7 Discussion

### 7.1 Permafrost aggradation

Few studies have examined the relationship between ground ice characteristics and post-glacial landscape development in Greenland (Gilbert et al., 2016). Pollard and Bell (1998), however, presented a model for ground-ice aggradation within epigenetic permafrost below the Holocene marine limit in the Eureka Sound lowlands on the Fosheim Peninsula, in high Arctic Canada. Here, the nature and distribution of ground ice was related to sea-level change, sediment grain-size and the mode of permafrost aggradation. Massive intrasedimental ice bodies were observed at the contact between marine silts and underlying gravels and sands, deposited during transgression. Low hydraulic conductivities in the overlying fine-grained sediments did not permit sufficient water migration resulting in the formation of reticulate, layered, lenticular, and pore cryofacies (Pollard, 2000a, b). Massive ice bodies and cryofacies were interpreted to have formed during and following deglaciation, when glacier melt water and brackish sea water were available to recharge local groundwater systems.

C1 and C2 are generally ice-poor and the majority of the cores are characterized by the pore cryofacies. Cryofacies with visible ice (La and Su) are restricted the fjord-basin and delta-slope facies associations. A similar pattern is observed in gravimetric moisture content and excess ice content (Figure 7; Figure 8). The distribution of ground ice in the Zackenberg delta is similar to that described by Pollard (2000a, 2000b). The presence of the layer cryofacies in epigenetic permafrost has also been described by Kanevskiy et al. (2014) in lacustrine sediments in interior Alaska. The suspended cryofacies is likely a more developed form of the layered cryofacies, forming during increased moisture availability or slower rates of freezing. The absence of appreciable ground ice in the remainder of the core indicates that permafrost in the Zackenberg lowlands is primarily epigenetic – forming following sea-level fall during the early Holocene. During permafrost aggradation, ground-water was likely recharged by glacier meltwater or the incursion of sea-water.





The presence of syngenetic permafrost is inferred using a combination of sedimentology and the dating results in the top of C2 – which record continued sedimentation following subaerial exposure. Here, the development of syngenetic permafrost is related to localized accumulation of niveoaeolian sediments and organic material (Christiansen, 1998b; Christiansen et al., 2002). The uppermost syngenetic component is almost three meters in thickness. Despite continuous

sedimentation and sufficient moisture availability, syngenetic permafrost at C2 consists of pore cryofacies. This is attributed to the coarse-grained sediment and the absence of frost-susceptible silts. Syngenetic permafrost is thus of local occurrence on the palaeo-delta surface and relates to periglacial, nival, fluvial and aeolian activity during the Holocene.

### 7.2 Valley-fill history and delta progradation

During the past 13 ka, the Zackenberg Lowlands have undergone major environmental changes through periods of glacial,
marine, fluvial and periglacial dominance. Sedimentological, cryostratigraphic and geomorphological observations are combined with the OSL dating results to produce a model for the landscape and permafrost development of the Zackenberg Lowlands. This model is condensed into three stages – illustrating the major changes in sediment supply, sea level and sedimentary processes following glaciation (Figure 9). The model provides a framework in which to discuss changes in the sedimentary facies and cryostratigraphy, and improves our understanding of the amount of ground-ice in Arctic valleys. This
is important as the amount of ground ice determines the potential landscape response due to climate change.

A stratigraphic model for fjord-valley infilling in Norway has been presented by Corner (2006). This tripartite model consists of a deglacial transgressive systems tract (DTST), a deglacial highstand systems tract (DHST), and a postglacial forced regressive systems tract (PRST). The DTST is deposited during deglaciation by retrogradational stacking of proglacial sediments. Sedimentation is concurrent with marine inundation and the early stages of sea-level fall during
isostatic rebound. The DHST is characterized by the rapid progradation of glaciofluvial deltas and concurrent deposition of glaciomarine deposits on the basin floor. Lastly, the PRST forms following areal deglaciation and is characterized by fluvial delta progradation, fluvial incision, and terracing during relative sea-level fall. A similar model was developed for the Scoresby Sound region, ca. 750 km south of the Zackenberg area by Hansen (2004). Though the details of the stratigraphy vary between locations, the underling controls identified in these models – sediment supply, accommodation space, and
relative sea-level change – remain the same.

### 7.2.1 Deglaciation and sea-level highstand

The regional glaciation during the LGM is recorded by a presence of a basal till in FA I (Christiansen and Humlum, 1993; Christiansen et al., 2008). Though earlier studies have suggested a deglaciation age of 11.7 to 10.1 ka (Bennike and Weidick, 2001; Bennike et al., 2008), the results of this investigation indicate that the Zackenberg lowlands were ice-free by between
13 ka and 12 ka (Table 2; Figure 7 and 8). This discrepancy may be because previous studies have relied on radiocarbon dating of marine macrofossils. Environmental restrictions on the organisms which produce these shells such as brackish water, high-sedimentation rates and suspended sediment concentrations may have restricted their presence during



deglaciation (Netto et al., 2012). Ice retreated initially to a grounding line at the mouth of the Zackenberg Valley, depositing an end-moraine complex (Figure 1C; Figure 9A).

Following deglaciation of the Young Sound, the Zackenberg lowlands were inundated by the sea. The maximum flooding surface corresponds with the transition from glacial (FA I) to glaciomarine (FA II) deposits. This stratigraphic

boundary marks the maximum landward incursion of marine-influenced deposits following deglaciation (Hansen, 2004). At Zackenberg, the transition from the DTST to DHST is only observed at three locations (S6, C1 and C2) and is marked at the transition from upwards fining to upwards coarsening in FA II (Figure **5**). Marchand et al. (2013) observed a similar transition in the deposits of the Matane River Valley (Quebec, Canada), noting a reduction in ice-rafted debris in the non-glacially influenced facies. The scarcity of drop-stones in most of FA II and FA III suggests that areal deglaciation started

shortly after the transition to FA II. The OSL dates from C1 and C2 suggest that this transition was approximately 12.5 ka.

During the DHST, the Zackenberg basin received sediment from suspension plumes and low-density turbidity currents contributing to the aggradation of FA II. A shallow water high-stand delta likely developed in front of a proglacial melt-water plain during this period. Deltaic deposits were not observed north of C1. Permafrost was likely absent in the lowlands during the DTST and DHST as thermal boundary conditions in the marine environment and beneath the unstable braided-

river channels were too warm to permit permafrost aggradation.

### 7.2.2 Delta progradation and relative sea-level fall

Relative sea-level fell during the Holocene and reached its present day limit ca. 4.5 ka (Bennike et al., 2008; Pedersen et al., 2011). The topography of the glacial and glaciofluvial landscape deposited during the DTST and DHST controlled the delta location. To the east, the adjacent lowlands (ca. 30-40 m a.s.l.) are higher in elevation than the pre-delta terrain surface. The

delta was thus deposited in the lowest part of the landscape. During sea-level recession, river networks were directed to the west of the glacial and glaciofluvial deposits, where the deltaic deposits are located today (Figure 1C; Figure 9B). The OSL ages from the upper part of the delta in C1 and C2 constrain the timing of initial delta deposition to between 13 and 11 ka (Table 2). Christiansen et al. (2002) presented dating results indicating that the delta was active at S6, 9.5 ka.

Most of the sediments comprising FA II, FA III, and FA IV were deposited during the PRST. During this period,

relative-sea level was in decline resulting in a reduction in accommodation space in the basin. Areal deglaciation reduced fluvial discharge and the available sediment to that which could be mobilized and reworked by the palaeo-river system (Christiansen and Humlum, 1993). Progradation of the delta continued during the PRST, but the rate of progradation was probably slower than during the DHST, due to the factors mentioned above (Corner, 2006; Eilertsen et al., 2006). Despite the declining accommodation space, the sedimentary facies of FA III at the fjord-proximal sites (S1-S6) reflect deeper water

conditions, as the delta prograded into the basin (Eilertsen et al., 2011).

The rapid incision of the Zackenberg River during Holocene sea-level decline limited the erosion of the underlying facies, forming the terraces and exposures documented in this study. FA IV records variable hydraulic conditions in the palaeo-Zackenberg River. At S9, FA IV was deposited at part of a large glaciofluvial outwash plain, where sediment was



primarily transported as bedload. The river switched to a single-channel system with mixed sediment transport once incision began ca. 9.5 ka. This could be explained by the exhaustion of the sediment source once the glacier retreated (Marchand et al., 2013).

Permafrost began to aggrade in the lowlands following sea level decline and delta progradation in the Late Weichselian
or Early Holocene. This is reflected in the assemblage of cryofacies, indicating epigenetic permafrost formation. The formation of the La and Su cryofacies in FA II and FA III was likely facilitated by glacier melt water or seawater incursion that provided a moisture source for excess-ice development. Permafrost landforms, such as ice-wedge polygons, podsols also began to form during this time (Christiansen et al., 2002; Christiansen et al., 2008).

### 7.2.3 Present delta system

The final stage of delta formation encompasses the period of relatively stable sea-level after 4.5 ka. Landscape changes after this time were minor compared to those during sea-level fall. The raised terraces reflect shifting zones of erosion and deposition during the PRST and Holocene emergence. Progradation rate of the delta declined as availability was restricted and the delta approached the edge of the Young Sound. Sediments in the drainage basin consists mainly of pergiglacial and glaciofluvial deposits. Late Weichselian glacial deposits have primarily been reworked (Figure 9C). Deep incision during
relative sea-level fall limited erosion, preserving the terrace morphology.

Ice-wedge polygons formed and periglacial nivation processes have been active in the delta since its exposure as permafrost probably established quickly in this high-Arctic setting (Christiansen et al., 2002). Also, palsas developed in the low-laying areas since emergence. In addition, niveo-fluvial and niveo-eolian sediment transport continues to modify the landscape downslope of snow patches (Christiansen, 1998b; Christiansen et al., 2002). At present, permafrost exists under all
terrestrial surfaces.

### 7.3 Application to other formerly glaciated valleys

Recent studies have presented models for infilling of formerly glaciated fjord-valley from Greenland (Hansen, 2001, 2004), Norway (Nemec et al., 1999; Corner, 2006; Eilertsen et al., 2006; Hansen et al., 2009; Eilertsen et al., 2011), Canada (Marchand et al., 2013), Sweden (Plink-Björklund and Ronnert, 1999), and Germany (Winsemann et al., 2007). These
investigations demonstrate that fjord-valleys are primarily infilled during deglaciation and relative sea-level fall, resulting in a complex stratigraphy, which can vary substantially between locations. However, recent studies also indicate that the underlying depositional regime controls are the same (Eilertsen et al., 2011). With the exception of the investigations in Greenland, these studies are restricted to non-permafrost environments. No previous studies have investigated ground ice or cryofacies in valley-fill deposits.
Previous studies have identified similar infilling patterns of valleys following regional deglaciation. Given similarities in sedimentary facies and landscape development, it is reasonable to expect that the aggradational history of permafrost is similar as well. This suggests that permafrost below the upper marine limit of large tributaries valleys throughout northeast



Greenland is likely a Holocene phenomenon, with the exact age corresponding to the timing of subaerial exposure following marine transgression. These coarse-grained deposits contain an assemblage of cryofacies characteristics of ice-poor epigenetic permafrost.

## 8 Summary and conclusions

The valley-fill deposits in the Zackenberg lowlands formed during highstand and RSL fall following deglaciation, 12 ka to 11 ka. The majority of the sedimentary deposits accumulated by the early Holocene, ca. 10 ka. The reduction in accommodation space during relative sea-level fall and high glacial and paraglacial sediment yield resulted in rapid sedimentation and progradation of the delta. Permafrost began to aggrade in subaerial land surfaces following exposure, ca. 11 ka. In the lowlands, permafrost history is closely tied with delta progradation.

The following conclusions are drawn from this study:

- Following deglaciation, three distinct phases of delta formation are recognized in the sedimentary facies and cryofacies. The majority of the delta accumulated during the late Weichselian and early Holocene. Rapid sedimentation and delta progradation is attributed to high sediment yield from glaciers and glaciofluvial erosion and transport and the decline in accommodation space during sea-level fall.

- The OSL ages suggest that the Zackenberg area was deglaciated prior to ca. 12.5 ka. This is 2 ka earlier than previously reported AMS [14]C ages. The discrepancy between these methods is attributed to the high-sedimentation rates and suspended sediment in the fjord during deglaciation, which limited the establishment of marine macrofauna.

- Permafrost in the Zackenberg lowlands is a Holocene phenomenon. The vertical distribution of cryofacies and absence of appreciable ground ice in frost-susceptible sediments indicates permafrost in the Zackenberg Delta deposits post-dates deglaciation, 11 ka. The onset of conditions conducive to permafrost aggradation is concurrent with subaerial exposure following RSL decline or delta progradation. The resultant epigenetic permafrost is ice-poor, overall. The results of this investigation may have applications for other formerly glaciated valleys in permafrost regions.



**Acknowledgements**

This investigation was financed by the PAGE21 (Changing permafrost in the Arctic and its Global Effects in the 21[st] Century) project – grant agreement number 292700 a seventh framework EU programme. Additional funding was provided by the Centre for Permafrost (CENPERM) at the University of Copenhagen, funded by the Danish National Research

5    Foundation (CENPERM DNRF1000) and by the Nordic Centre of Excellence, DEFROST (Impacts of a changing cryosphere – depicting ecosystem-climate feedbacks from permafrost, snow and ice). Christine Thiel received funding from the German Research Foundation (DFG grant TH1651/1-1). We gratefully acknowledge the hospitality and assistance of the staff with drilling at the Zackenberg Ecological Research station. Special thanks to Ulrich Neumann (Kolibri Geoservices) and Jordan Mertes for their assistance during drilling in summer 2012.



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



**Table 1. Sedimentary facies descriptions and interpretations.**

| FACIES | FACIES DESCRIPTION | BED CONTACTS & THICKNESS | INTERPRETATION |
|---|---|---|---|
| 1. Diamicton (Figure 3) | Sandy, matrix supported diamicton with clast supported portions. Over-consolidated compared to other units | Lower contact is not observed, upper contact is sharp. 2-6 m thick in sections and cores. | Basal till – based on compaction, clast contact, position and extent (Christiansen and Humlum, 1993; Benn and Evans, 2010). |
| 2. Silts (Figure 4) | Weakly laminated silts with variable degrees of bioturbation. | Sharp or transitional contacts. Up to 1 m thick. | Fallout from suspended sediment plumes and deposition by mud-rich turbidity currents (Hansen, 2004). |
| 3. Interlaminated sands and silts (Figure 4) | Fine-sands and silts with occasional outsized clasts. Individual lamina range from 2 to 20 mm in thickness and may be either graded or ungraded. | Lower boundary is commonly diffuse while upper contact may be sharp. Beds up to 50 cm thick. | Deposition by turbidity currents and suspended sediment fallout (Reading, 2009). |
| 4. Normally graded sands and silts (Figure 4) | Normally graded beds of sand and silt. Some beds fine upwards into planar-parallel lamination or ripple cross-laminated sands or silts. | Lower boundary is sharp and may contain evidence of loading, scour, or water escape. Upper boundary is gradational or sharp. Beds are 1 to 25 cm thick. | Deposition by surge-like turbidity currents (Plink-Björklund and Ronnert, 1999). |
| 5. Cross-stratified sands (Figure 3) | Fine to coarse grained sand with planar cross-stratification | Sharp, erosive base, sharp upper contact. Beds are 10 to 40 cm thick. | Deposition by unidirectional, tractional currents. Cross-strata record the migration of dunes (Winsemann et al., 2007). |
| 6. Laminated sands (Figure 4) | Laminated to diffusely laminated very-fine to coarse sands. | Sharp lower contact. Sharp or gradational upper contact. Beds up to 20 cm thick. | Deposition from sustained hyperpycnal currents subject to waxing and waning (Hansen, 2004). Alternatively, upper-flow regime planar-parallel stratification (Reading, 2009). |
| 7. Massive pebbly sands (Figure 3) | Matrix-supported pebbly sands. Massive or weak-inverse grading. Clasts are granule to pebble-sized. | Sharp upper and lower contacts. Beds are 5 to 50 cm thick. | Deposition from sandy debris flows (Nemec, 1990). Either as Gilbert-type delta foresets or fluvial channel deposits. |
| 8. Stratified pebbly sands and gravels (Figure 3) | Clast-supported gravels and pebbly sands, high-angle cross-bedding (up to ca. 20°). | Sharp, flat upper and lower contacts. Beds are 10 to 50 cm thick. | Foreset beds deposited from grain-flows or noncohesive debris flows (Plink-Björklund and Ronnert, 1999; Hansen, 2004). |
| 9. Gravels (Figure 3) | Clast-supported massive or diffusely stratified gravels, with evidence of imbrication. | Sharp, flat, occasionally erosive contacts. Beds are 30 to 120 cm thick. | Deposited by tractional currents (Miall, 2010). Likely bedload transport in a braided-river system. |





**Table 2. Summary of core samples for luminescence dating, depths, quartz OSL and feldspar IRSL equivalent doses (De) and ages. The feldspar doses and ages are based on the measurements of three aliquots per sample and are only used to check for incomplete signal resetting. The quartz ages are the basis for the chronologies of the sediment cores. n = number of aliquots. s.e. = standard error.**

| Site | Laboratory ID | Depth [m] | Quartz OSL | | | Feldspar ISRL | | | |
|------|---------------|-----------|------------|------------|------------|---------------|---------------|------------|------------|
| | | | n | De ± s.e. [Gy] | age ± s.e. [ka] | $IR_{50}$ De ± s.e. [Gy] | $pIRIR_{225}$ De ± s.e. [Gy] | $IR_{50}$ age ± s.e. [ka] | $pIRIR_{225}$ age ± s.e. [ka] |
| C1 | 131543 | 0.74 | 20 | 29.8 ± 1.1 | 11.9 ± 0.7 | 46 ± 5 | 103 ± 9 | 15 ± 2 | 34 ± 3 |
| | 131544 | 1.67 | 18 | 32.1 ± 0.7 | 11.9 ± 1.2 | 58 ± 2 | 156 ± 9 | 18 ± 2 | 48 ± 5 |
| | 131547 | 4.61 | 18 | 35 ± 2 | 12.6 ± 0.8 | 55.8 ± 0.8 | 145 ± 6 | 16.6 ± 0.7 | 43 ± 3 |
| | 131548 | 5.19 | 18 | 45 ± 2 | 12.4 ± 0.7 | 56 ± 4 | 155 ± 8 | 13.4 ± 1.1 | 37 ± 3 |
| | 131550 | 7.05 | 20 | 47± 4 | 12.9 ± 1.3 | 80 ± 9 | 205 ± 32 | 19 ± 2 | 49 ± 8 |
| | 131552 | 8.31 | 20 | 38 ± 4 | 13 ± 2 | 67 ± 4 | 175 ± 12 | 19 ± 2 | 50 ± 6 |
| C2 | 131554 | 0.91 | 18 | 4.3 ± 0.2 | 1.6 ± 0.1 | 3.7 ± 0.2 | 8.6 ± 0.1 | 1.2 ± 0.1 | 2.7 ± 0.1 |
| | 131555 | 1.15 | 18 | 14.9 ± 1.1 | 7.4 ± 0.7 | 14.8 ± 0.3 | 22 ± 4 | 5.8 ± 0.3 | 9 ± 2 |
| | 131556 | 1.70 | 18 | 23.9 ± 0.7 | 9.7 ± 0.5 | 20 ± 2 | 41 ± 7 | 6.5 ± 0.6 | 14 ± 2 |
| | 131557 | 1.94 | 18 | 25.8 ± 1.0 | 10.3 ± 0.6 | 20.4 ± 0.8 | 41 ± 2 | 6.7 ± 0.4 | 13.4 ± 0.9 |
| | 131558 | 2.67 | 18 | 26.6 ± 1.3 | 11.8 ± 0.8 | 28.9 ± 0.6 | 85 ± 8 | 10.3 ± 0.5 | 30 ± 3 |
| | 131560 | 4.33 | 18 | 31.7 ± 1.0 | 11.7 ± 0.6 | 38 ± 2 | 115 ± 7 | 11.6 ± 0.7 | 36 ± 3 |
| | 131561 | 9.34 | 18 | 25.9 ± 1.0 | 10.6 ± 0.6 | 34 ± 3 | 107 ± 9 | 11.4 ± 1.0 | 36 ± 3 |
| | 131562 | 11.52 | 18 | 39.2 ± 1.0 | 12.0 ± 0.7 | 39.0 ± 0.9 | 126 ± 6 | 10.3 ± 0.5 | 33 ± 2 |





**Figure 1. A) Location of the study region in northeast Greenland. B) Regional map of central Northeast Greenland showing the location of the study area and key locations mentioned in the text. C) Details of the study area in the Zackenberg Lowlands. The glacial and glaciofluvial landscape formed during the late Weichselian, whereas the delta terraces are of late Weichselian to Holocene age. The location of sections (S1-S9) are denoted by black dots. Sediment core locations (C1 & C2) are denoted by white dots.**





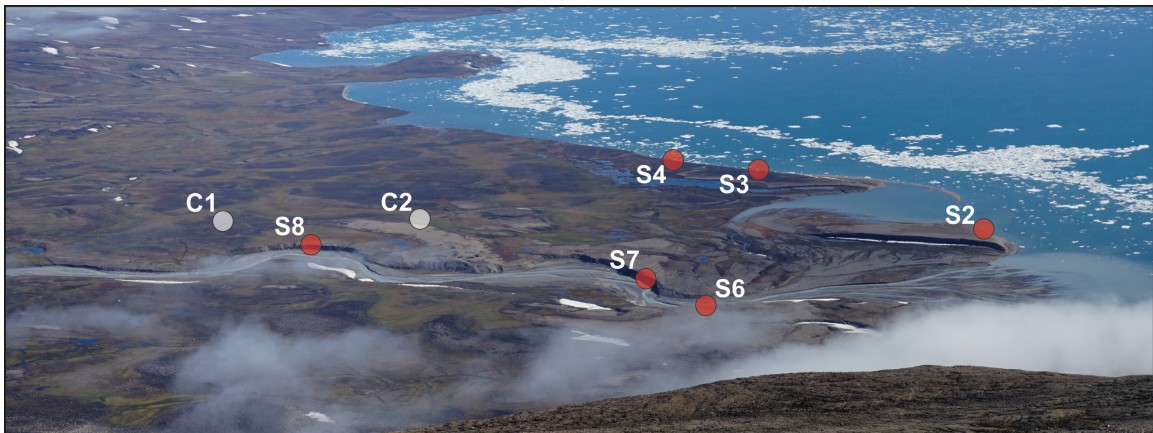

**Figure 2. Overview image from the Zackenberg Mountain looking east over the Zackenberg Lowlands. Site locations are those indicated in Figure 1. The photo was taken on 30 August 2015.**





**Figure 3.** Coarse-grained sedimentary facies. (A) cross-stratified sands (facies 5) and massive pebbly sands (facies 7). Image from S3 at 6 m a.s.l. (B) massive gravels (facies 9) and massive pebbly sands (facies 7) at S9 (41 m a.s.l.). Note trowel is 30 cm in length. (C) Diamicton (facies 1) – the lowermost exposed unit at S7 (ca. 14 m a.sl.). (D) Cross-stratified gravels and pebbly sands (facies 8) at S6 (ca. 27 m a.s.l.). Individual strata are separated by dashed lines.



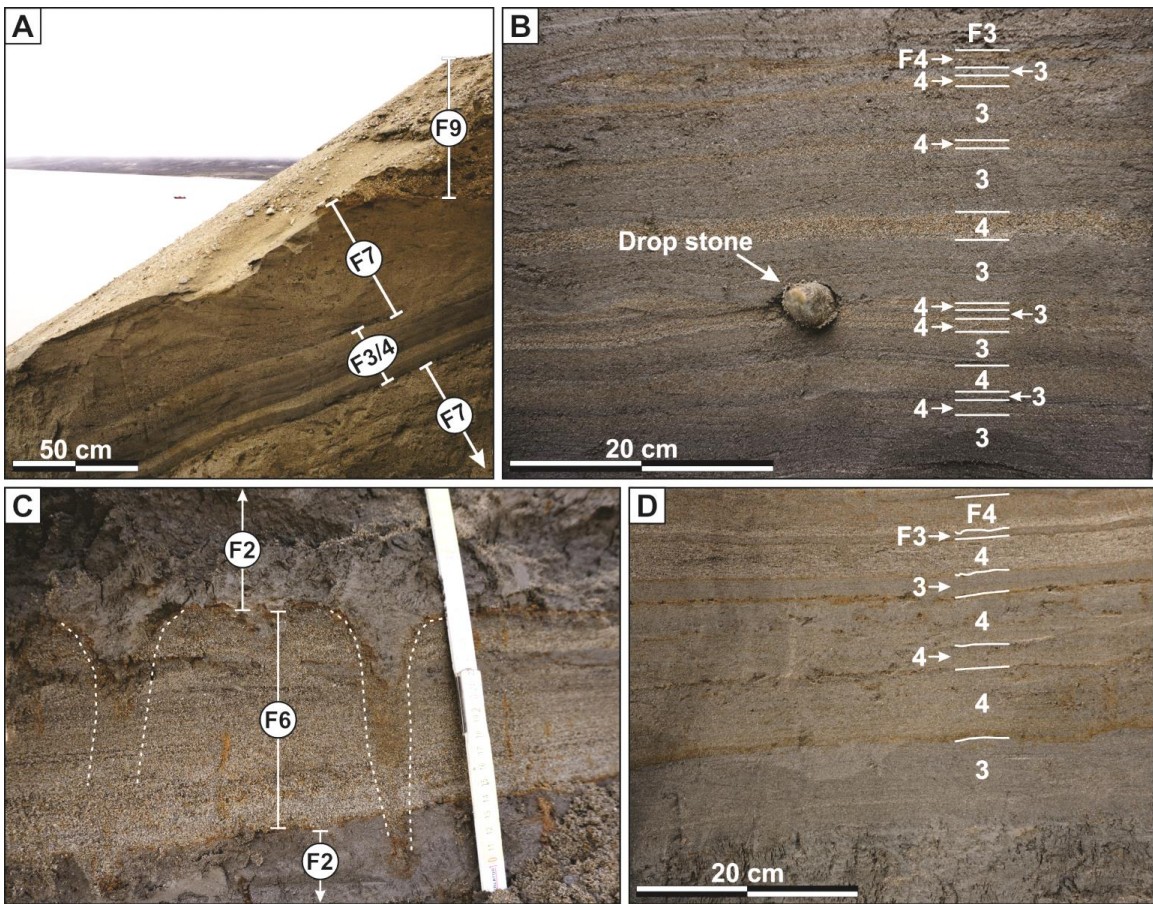

Figure 4. Fine-grained sedimentary facies. (A) Massive gravels (facies 9) truncating massive pebbly sand (facies 7), interbedded sands and silts (facies 3) and graded sands (facies 4) at S2 at 11 m a.s.l. (B) Alternating facies 3 and facies 4 at S5 (9 m a.s.l.). Note the presence of an outsized clast. (C) Bioturbated silts (facies 2) and laminated sands (facies 6) at S5 (6 m a.s.l.). Note vertically-oriented burrows outlined with white-dashed lines. (D) Alternating facies 3 and facies 4 at S2 (5 m a.s.l.).





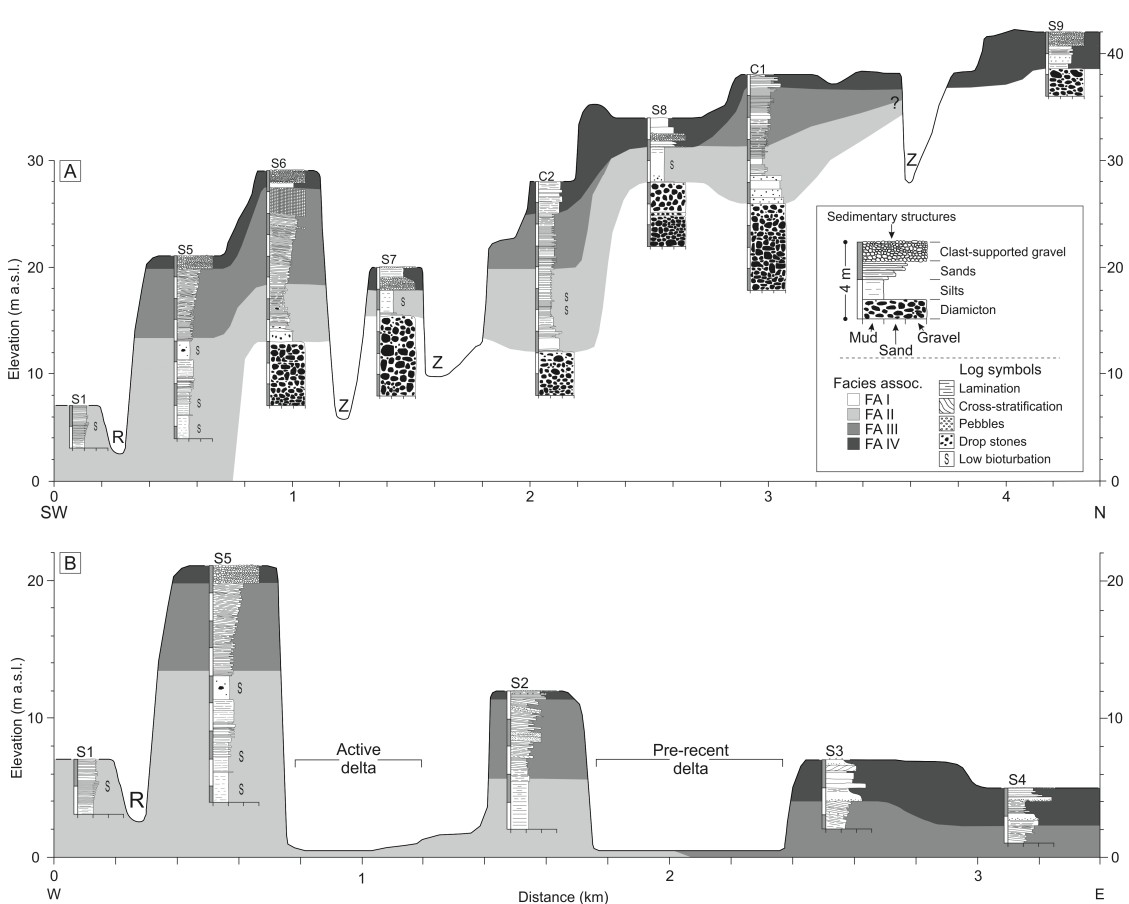

**Figure 5.** Correlation panel of sedimentary logs and facies associations from (A) the sections along the Zackenberg River (S1 & S5-S9) and cores (C1 & C2) and (B) coastal section (S1-S5). The site identities, above each log, correspond to those in Figure 1. Z and R indicate the location of the Zackenberg River and minor streams, respectively.



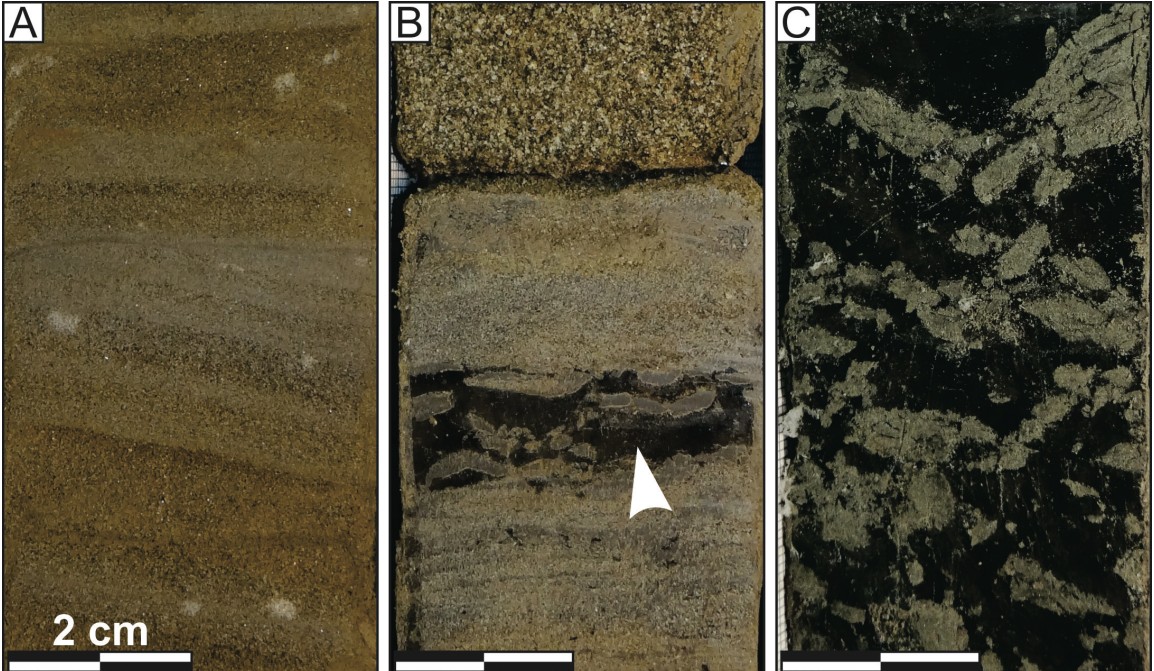

**Figure 6. Cryofacies. (A) pore (Po) cryofacies. (B) layer (La) cryofacies indicated by the white arrow; (C) suspended (Su) cryofacies. Note the ice appears black in the images.**





**Figure 7.** Log for C1 illustrating vertical variations in sediment grain size, facies associations, ground-ice characteristics and OSL ages. D, Po, La, and Su denote disturbed sections, pore, layered, and suspended cryofacies, respectively.



**Figure 8.** Log for C2 illustrating vertical variations in sediment grain size, facies associations, ground-ice characteristics and OSL ages. D, Po, and La denote disturbed sections, pore, and layered cryofacies, respectively. Note the difference in axis range compared with Figure 7.





**Figure 9. Schematic model of the three main stages of landscape development of the Zackenberg delta and adjacent lowlands. The model illustrates the essential processes and sediment sources which contributed to the development of the valley-fill deposits. A. Late-Weichselian deglaciation and inundation. Sediment was supplied by a proximal glacier and by mobilization of glacial deposits by contemporary fluvial networks. B. Early-Holocene sea-level fall and areal deglaciation. High sedimentation rates and rapid delta progradation during sea-level fall. Sediment supplied primarily through the reworking of deposits by glaciofluvial erosion and incision during isostatic uplift. The landscape is slowly dissected by a meandering braided-river system resulting in the terracing observed in the landscape today. Locally, permafrost began to aggrade in the lowlands following subaerial exposure. C. Modern landscape with select sections and coring locations. Sediment supplied by the fluvial reworking of raised deposits. Permafrost is continuous under stable, exposed land surfaces.**