# Peer review of "Cryostratigraphy, sedimentology and the late Quaternary evolution of the Zackenberg Delta, Northeast Greenland"

_The Cryosphere, 2016_

## Referee Comment (RC1) · Anonymous Referee #1 · 21 Feb 2017

General comments The paper includes a detailed facies description of the Zackenberg Delta in North-eastern Greenland. This based mostly of cryostratigraphic description and correlations, supported by sediment observation and analyses and geochronology. It is a solid work, which reveals further parts of the deglaciation history of Northeast Greenland.

Specific comments The abstract seems to be quite short and general. It is more a description of the Zackenberg Delta landscape than a summary of the objectives, methods and conclusion from the paper. I would like to recommend to rewrite and to extend the abstract. Grain size measurements are mentioned in the methods (Measurements included descriptions of sediment type, grain size, sorting . . .) but it was nothing written

about the analyses. It would be important to know, how the grain sizes were determined. The geochronology is given too little consideration, both in methods (only four lines) and in results (chapter 6, Table 2). It would be good to extend the explanation in the methods and the interpretation in the result chapter. More than halve of Table 2 contain data from IRSL feldspar measurement, with different abbreviation (IR50, pIRIR225). But nothing is mentioned in the method, result and discussion chapters.

Technical corrections p3 l15. p4 l16, p12 l5 early Holocene p3 l26, p10 l9, l11, Figure 1 and 2 caption Zackenberg lowlands p5 l13 a system adapted from Murton, (2013) and (French and Shur, (2010). p8 l9/10 The cryofacies abbreviation (Po, La, Su and D) should be named when the facies are firstly mentioned. p11 l11 the Zackenberg Basin p12 l4 in the late Weichselian p12 l6/7 Something seem to be missing in this sentence: Permafrost landforms, such as ice-wedge polygons, podsols also began to form during this time p12 l22 . . . of formerly glaciated fjord-valleys from . . . Table 2 The abbreviation for the feldspar-IRSL-dating are not explained

---

## Referee Comment (RC2) · Anonymous Referee #2 · 24 Feb 2017

General Comments: An interesting paper that continues to mine data from a geographical area of focussed research and environmental monitoring spanning several decades. Authors attempt to integrate a wide scope of studies based on coring of two raised delta terraces, linking sedimentology with cryostratigraphy

Specific Comments: Although not necessarily a main focus of the paper, the chronological story, for which their OSL data is applied is somewhat muddled. It has to be acknowledged that the RSL curve for this area is poorly constrained in the immediate period following deglaciation, when ∼half of the total uplift (∼40 m) occurs (at suggested rates of 10 mm/a). The 10.1 calibrated AMS age on bottom-set beds (∼31 m asl) must therefore be recognized simply as a minimum estimate for deglaciation,

and in the absence of sedimentological evidence where one can trace bottom-sets into foresets and potentially topsets, it is not possible to so precisely ascribed the paleo-sea-level to which they are accordant. Thus, projecting the upward part of the RSL curve on a sample with a wide potential paleo-depth level is problematic (the M. truncata species itself potentially occupies a wide habitat depth). It is not surprising then that OSL ages could be older than this radiocarbon (calibrated) date. It becomes problematic, however, in that potentially this implies that they have foreset/topset beds that are older than what may well be accordant bottomset beds. If the foreset beds are dating 12-13 ka...then this would require and even earlier deglaciation, as the entire period of post-glacial sea level fall between marine limit (70 m asl) and their uppermost delta (38 m asl) would have to occur before the sediments that are forming these two deltas aggrade. Why is incomplete bleaching of sediments that would have been part of a very turbid, potentially short transport distance, not more considered/discussed? Authors mention the inclusion of feldspar ISRL as a check on this, but do not discuss the results within the paper. There is no reason to suspect the C14 ages are wrong, other than perhaps generally small regional variations in marine reservoir correction. What needs to be better qualified by them is the stratigraphic relationship between their OSL ages and those on marine macrofossils.

Facies recognition from core samples only 42mm in diameter is difficult, at best. While detailed architecture and facies descriptions were based off of river-cut bluff exposures at "sections," it would have helped if at least one of the drill holes was established in direct proximity to an existing section in order to better tie the core and section logs together.

Given comparable sedimentology between C1 and C2...why is the moisture content of C1 generally higher, and also, what is the explanation for why there are more significant excursions in higher moisture content and excess ice content? Would this not, in part, reflect lateral variations in sedimentology unresolvable at the core level - that is, it suggests that while both deltas are fairly coarse in nature, and thus would have largely

been free-draining, that because of lateral variations in sediment texture/sediment homogeneity, may have permitted more effective fluid permeabilities? While I understand the valley-fill history model, I question it's application to an environment that is more akin to broad lowland.

Why is atmospheric moisture-sourced groundwater eliminated as a potential source of the epigenetically formed ground/pore ice? Depending on the configuration of the retreating ice, much of the surrounding terrain could have been subaerially exposed, contributing to regional groundwater tables. In the absence of chemical/isotopic study, I'm uncertain how the authors can simply state the interstitial water was glacial meltwater/marine sourced. Authors also do not discuss the implications of saline permafrost, and the role this may have had on enhanced drainage of coarse sediments as the depressed freezing point may have perpetuated fluid flow through the sediments during permafrost aggradation...would this have implications on cryostratigraphy?

Presumably part of the importance of understanding/reconstructing the cryostratigraphy is to enable some comment about ice content (p10L14 - "understanding of the amount of ground-ice in Arctic valleys) vis a vis potential melt volume/sediment compaction were this to thaw...can this be meaningfully done with your data?

Technical Comments: p3L20 - as previously discussed...the 9.5 ka shell date is a poor constraint on the 70 m marine limit and it's projection to 10.1 ka p6L30...what are the diagnostic sedimentological properties of a "dilute" turbidity current? p7L7 - a lack of IRD clasts does not have to reflect high sedimentation rates and dilution - it could simply be an absence of IRD reflecting lack of calving margins; it could also reflect that the delta currents move debris-laden ice off-shore, so little chance to accumulate. p9L5 - raised deltas would usually be isolated pretty quickly after emergence, so I'm unsure how you see these continuing to aggrade through the Holocene in any kind of fluvial sense...presumably Aeolian activity could be responsible? Table 2 - while I know you can mine the data out of Figs 7/8, it would be helpful if you included a column in this table that showed the FA interp for each of the samples

---

## Referee Comment (RC3) · M. Fritz (Referee) · 14 Mar 2017

The manuscript provided by Gilbert et al. seeks to reconstruct environmental conditions that determine sediment facies development and permafrost characteristics in a proglacial and periglacial environment of the high Arctic along the northeast Greenland coast. This study is especially interesting because it provides baseline geoscientific data on postglacial landscape development of an area that is a long-term monitoring station in the Arctic. Any kind of study relating to e.g. landscape formation, soil sciences, carbon turnover, microbiology, vegetation succession, sea-level change, or coastal geomorphology will need such information on the timing of glacial meltdown, sedimentation milieu and permafrost development.

[Figure]

The authors present very thorough data and observations on sediment fabrics, grain-size composition and cryofacies in two sediments cores. Such small-scale observation is aided by similar data from natural exposures in the area. Fourteen OSL from the two cores ages serve as age control. Figures and tables are of excellent quality and help the general understanding of the manuscript. Overall the quality of the manuscript is very good, the language is clear and almost free of mistakes.

I suggest the manuscript to be accepted after minor revisions.
* * *
General comments:

1. The abstract is rather short and could benefit from more results and interpretation so that the main message of the manuscript comes across right with the abstract.

2. OSL Dating Please provide more detail on the OSL dating in the methods, especially how you have dealt with potential errors such as fading or incomplete bleaching. You should also provide reasoning (not only in the figure caption) why both OSL on quartz and IRSL on feldspar were measured but only OSL ages are discussed. One major novel statement of the manuscript is the presumably earlier recession of the ice from the study area based on OSL ages in comparison to earlier studies based on 14C radiocarbon ages. OSL ages usually come with higher uncertainities and standard errors than 14C-ages in this age range. However, the errors are not discussed here although they could easily move your OSL ages in the same age range as obtained by Bennike. Bennike's ages are right at the termination or after the termination of the Younger Dryas. In contrast, your ages are right within the Younger Dryas which is not prominent for widespread deglaciation but for the opposite. Please re-consider your arguments.

3. The authors use cryofacies terms that do not exist as such. (1) pore cryofacies, (2) layer cryofacies both are not cryofacies terms as it can be found or related to the

literature. Pore ice is a certain ground ice type. Better use the term "structureless" as introduced in Murton and French 1994 and also used in French and Shur 2010. Then replace the new term throughout the manuscript. The same applies to "layer cryofacies" that should be replaced throughout by "layered". Be careful to also change this in the figure captions.
* * *
Specific comments:

For specific comments see the annotated and attached pdf-file.

Michael Fritz (Alfred Wegener Institute, Helmholtz Centre for Polar and Marine Research)

Please also note the supplement to this comment:
http://www.the-cryosphere-discuss.net/tc-2016-299/tc-2016-299-RC3-supplement.pdf

**Supplement:**

[revised manuscript text omitted]

---

## Author Comment (AC1) · 7 Apr 2017

We thank the referee for their constructive critique of this manuscript. The referee's comments and suggestions have been very helpful in improving this manuscript.

**Specific comments**

1. Referee comment: "The abstract seems to be quite short and general. It is more a description of the Zackenberg Delta landscape than a summary of the objectives, methods and conclusion from the paper. I would like to recommend to rewrite and to extend the abstract."

*Response: We thank the reviewer for their comment. We have revised the abstract.*

2. "Grain size measurements are mentioned in the methods (Measurements included descriptions of sediment type, grain size, sorting...) but it was nothing written about the analyses. It would be important to know, how the grain sizes were determined."

*Response: Samples were taken from the cores and analyzed in the laboratory using a combination of sieve and laser particle-size analysis. In the field (sections), a hand-lens was used to visually determine the dominant grain-size class present (Tucker, 2011). As details of the grain-size distributions are not used in this investigation, we have simplified the methods section by stating that we performed sedimentary logging of the sections and cores.*

3. "The geochronology is given too little consideration, both in methods (only four lines) and in results (chapter 6, Table 2). It would be good to extend the explanation in the methods and the interpretation in the result chapter. More than halve of Table 2 contain data from IRSL feldspar measurement, with different abbreviation (IR50, pIRIR225). But nothing is mentioned in the method, result and discussion chapters."

*Response: Thank you for the comment. We have extended both the methodology and results sections concerning geochronology. Abbreviations are now fully explained as is the rationale for including IRSL information. We have applied IRSL (IR50 and pIRIR225) dates in combination with quartz-derived OSL ages to assess the bleaching of the sediments and thus the accuracy of the OSL ages. This is now explained in the results section of the manuscript.*

**Technical corrections**

1. p3 l15. p4 l16, p12 l5 early Holocene
*Response: done*
2. p3 l26, p10 l9, l11, Figure 1 and 2 caption Zackenberg lowlands
*Response: done*
3. p5 l13 a system adapted from Murton, (2013) and (French and Shur, (2010).
*Response: done*
4. p8 l9/10 The cryofacies abbreviation (Po, La, Su and D) should be named when the facies are firstly mentioned.
*Response: done*
5. p11 l11 the Zackenberg Basin
*Response: removed 'Zackenberg' …now reads as "…the basin…"*
6. p12 l4 in the late Weichselian
*Response: done*
7. p12 l6/7 Something seem to be missing in this sentence: Permafrost landforms, such as ice-wedge polygons, podsols also began to form during this time
*Response: the sentence is now corrected to read …polygons, and podsols…*
8. p12 l22 ...of formerly glaciated fjord-valleys from...

*Response: done*

9. Table 2 The abbreviation for the feldspar-IRSL-dating are not explained

*Response: We have corrected and defined abbreviations in Table 2.*

**References**

Tucker, M. E.: Sedimentary rocks in the field: a practical guide, John Wiley & Sons, 2011.

[revised manuscript text omitted]

---

## Author Comment (AC2) · 7 Apr 2017

We thank the referee for their detailed and thought provoking review. These comments have provided us with an opportunity to reconsider our initial interpretation of key background material and critically assess the inclusion of results from other publications. Furthermore, these comments have allowed us to clarify the OSL results and strengthen the cryostratigraphic interpretations.

**Specific comments**

1. Referee comment: "Although not necessarily a main focus of the paper, the chronological story, for which their OSL data is applied is somewhat muddled. It has to be acknowledged that the RSL curve for this area is poorly constrained in the immediate period following deglaciation, when ~half of the total uplift (~40 m) occurs (at suggested rates of 10 mm/a). The 10.1 calibrated AMS age on bottom-set beds (~31 m asl) must therefore be recognized simply as a minimum estimate for deglaciation, and in the absence of sedimentological evidence where one can trace bottom-sets into foresets and potentially topsets, it is not possible to so precisely ascribed the paleosea-level to which they are accordant. Thus, projecting the upward part of the RSL curve on a sample with a wide potential paleo-depth level is problematic (the M. truncata species itself potentially occupies a wide habitat depth). It is not surprising then that OSL ages could be older than this radiocarbon (calibrated) date. It becomes problematic, however, in that potentially this implies that they have foreset/topset beds that are older than what may well be accordant bottomset beds. If the foreset beds are dating 12-13 ka…then this would require and even earlier deglaciation, as the entire period of post-glacial sea level fall between marine limit (70 m asl) and their uppermost delta (38 m asl) would have to occur before the sediments that are forming these two deltas aggrade."

   *Response: We agree that sea level curve for the late Weichselian/early Holocene is poorly constrained. The curve presented by Bennike et al. (2008) extrapolates from the elevation of dated samples (ca. 31 m a.s.l.) to 70 m a.s.l. based on the geomorphological observations. In light of the reviewer's comment, chronostratigraphic evidence provided by Christiansen et al. (2002) and Pedersen et al. (2011), and the absence of any other regional evidence for marine limits approaching that elevation, we agree that 70 m a.s.l. is an unreasonable marine limit. We believe that the maximum terrace elevation of 38 m a.s.l. more accurately approximates the upper marine limit but recognize that this is again a minimum estimate. This fits with an upper marine limit of ca. 40 m a.s.l. reported by Christiansen et al. (2002). We have changed/corrected the background text accordingly.*

2. "Why is incomplete bleaching of sediments that would have been part of a very turbid, potentially short transport distance, not more considered/discussed? Authors mention the inclusion of feldspar ISRL as a check on this, but do not discuss the results within the paper. There is no reason to suspect the C14 ages are wrong, other than perhaps generally small regional variations in marine reservoir correction. What needs to be better qualified by them is the stratigraphic relationship between their OSL ages and those on marine macrofossils."

   *Response: Thank you for identifying this. We have updated the geochronology sections in the methodology and results to discuss incomplete bleaching and have expanded the results section to more clearly present the significance of the OSL and IRSL results. Several recent publications have focused on the use of OSL in arctic and glacial-proximal depositional environments (Fuchs and Owen, 2008; Alexanderson and Murray, 2012) and comparing OSL and infrared stimulated luminescence (IRSL) ages to assess bleaching (Murray et al., 2012). We have revised the manuscript to include these references and better explain our methodology and results. We do not believe the C14 ages are incorrect. The OSL results in this investigation simply augment the*

*geochronology from this region and provide an indication that deglaciation may have occurred earlier than previously supposed. We adjusted the interpretations in the manuscript to reflect this.*

3.  "Facies recognition from core samples only 42mm in diameter is difficult, at best. While detailed architecture and facies descriptions were based off of river-cut bluff exposures at "sections," it would have helped if at least one of the drill holes was established in direct proximity to an existing section in order to better tie the core and section logs together."

*Response: The two cores were retrieved within ca. 5 m of an existing 'road' network. Environmental restrictions at the Zackenberg Ecological Research Station made it impossible to transport this large drill rig to the section edges. We agree that sedimentary facies are best identified in outcrops and sections but note many geological studies develop facies models based on cores from unconsolidated sediment or bedrock. In this study we benefit from having the outcrop close to the boreholes and are able to tie together the stratigraphy observed in the outcrop with that in the cores.*

4.  "Given comparable sedimentology between C1 and C2...why is the moisture content of C1 generally higher, and also, what is the explanation for why there are more significant excursions in higher moisture content and excess ice content? Would this not, in part, reflect lateral variations in sedimentology unresolvable at the core level – that is, it suggests that while both deltas are fairly coarse in nature, and thus would have largely been free-draining, that because of lateral variations in sediment texture/sediment homogeneity, may have permitted more effective fluid permeabilities?"

*Response: Ground-ice content varies in response to three primary factors: sediment grain size, moisture availability (including ground-water hydrology), and the freezing history. As the reviewer has pointed out, sediment characteristics and the mode of permafrost formation (i.e. epigenetic) are similar at C1 and C2. The variation in ground ice content likely relates to either the rate of downward permafrost aggradation in relation to the position any potential moisture source or local variations in hydrology. As indicated by the referee, a likely culprit would be lateral variations in sediment properties and lithofacies and the effect this would have had on ground-water migration during permafrost aggradation. To account for this we have included the following sentence in the discussion: "The differences in ice content and cryofacies between C1 and C2 likely relate to either variations in moisture availability during permafrost aggradation at these sites or lateral variations in sediment characteristics that are unresolvable at the core scale."*

5.  "While I understand the valley-fill history model, I question it's application to an environment that is more akin to broad lowland."

*Response: We agree that the valley-fill model presented by Corner (2006) was primarily developed to explain sedimentary infilling in confined fjord valleys in Norway. However, the three regime controls identified by Corner – sediment supply during and following glacier retreat, basin-depth (accommodation space) variation, and glacio-isostatically driven sea-level change –control the post-glacial development of delta systems irrespective of the degree of horizontal confinement. For instance, Hansen (2004) presents a model of the deltaic infill of the Falsterselv region, also in east Greenland. The Falsterselv region had been previously described by Ingólfsson et al. (1994) as "extensive, undulating lowlands, transected by river channels and ravines". Hansen (2004) identifies relative sea level change, onshore and offshore topography (basin configuration), and sediment yield as the major regime controls during the deltaic infilling of this deglaciated*

*landscape. Other, similar examples can be found in the literature. As the regime controls at Zackenberg are fundamentally the same as in other deglaciated locations, we believe that these models provide a suitable foundation for explaining the sedimentary architecture observed in the Zackenberg Delta deposits.*

6. "Why is atmospheric moisture-sourced groundwater eliminated as a potential source of the epigenetically formed ground/pore ice? Depending on the configuration of the retreating ice, much of the surrounding terrain could have been subaerially exposed, contributing to regional groundwater tables. In the absence of chemical/isotopic study, I'm uncertain how the authors can simply state the interstitial water was glacial meltwater/marine sourced."

*Response: Thank you. We have removed "During permafrost aggradation, ground-water was likely recharged by glacier meltwater or the incursion of sea-water". We agree that concluding on the origin of water prior to freezing would require additionally analysis not conducted in this investigation.*

7. "Authors also do not discuss the implications of saline permafrost, and the role this may have had on enhanced drainage of coarse sediments as the depressed freezing point may have perpetuated fluid flow through the sediments during permafrost aggradation...would this have implications on cryostratigraphy?"

*Response: Solutes in the pore water would certainly depress the freezing point. This would increase the unfrozen water content for any given temperature and likely influence the temperature at which ground ice and cryofacies developed. Nevertheless, without unfrozen water content curves or information regarding the solute concentration in the pore water during permafrost aggradation it is difficult to speculate to what degree saline permafrost might impact cryostructures. This would make for an interesting laboratory investigation where the effects of pore-water salinity and temperature on ground-ice facies could be investigated in a controlled environment.*

8. "Presumably part of the importance of understanding/reconstructing the cryostratigraphy is to enable some comment about ice content (p10L14 - "understanding of the amount of ground-ice in Arctic valleys) vis a vis potential melt volume/sediment compaction were this to thaw...can this be meaningfully done with your data?"

*Response: Yes, we could estimate the amount of subsidence based on the excess ice content in the two cores. However, the aim of this manuscript is landscape reconstruction and not permafrost degradation. We would like to keep the manuscript focused. One of the key points from our investigation is that, with the formation of epigenetic permafrost in relatively coarse-grained deposits, the ice content in the valley bottom is quite low (albeit with variation within and between sites).*

**Technical corrections**
1. p3L20 - as previously discussed...the 9.5 ka shell date is a poor constraint on the 70 m marine limit and it's projection to 10.1 ka

*Response: We agree and, as elaborated on above, have changed the text accordingly.*

2. p6L30...what are the diagnostic sedimentological properties of a "dilute" turbidity current?

*Response: "dilute turbidity currents" has been replaced with "low-density turbidity currents". Diagnostic properties included an upwards fining reflecting decelerating flow speeds (Reading, 2009).*

3. p7L7 - a lack of IRD clasts does not have to reflect high sedimentation rates and dilution - it could simply be an absence of IRD reflecting lack of calving margins; it could also reflect that the delta currents move debris-laden ice off-shore, so little chance to accumulate.

*Response: We agree there are other factors which were not considered. We have amended the text by removing this argument.*

4. p9L5 - raised deltas would usually be isolated pretty quickly after emergence, so I'm unsure how you see these continuing to aggrade through the Holocene in any kind of fluvial sense...presumably Aeolian activity could be responsible?

*Response: Yes, post-emergence aggradation at C2 is attributed to niveo-aeolian activity (Christiansen, 1998). We have clarified this in the text.*

5. Table 2 - while I know you can mine the data out of Figs 7/8, it would be helpful if you included a column in this table that showed the FA interp for each of the samples

*Response: Thank you for this suggestion, we have included the FA for each sample in table 2.*

[revised manuscript text omitted]

---

## Author Comment (AC3) · 7 Apr 2017

**Referee #3: Dr. Michael Fritz**

Thank you very much for your constructive review. These have been instructive and helpful in improving the clarity and quality of this manuscript.

**General comments**

1. Referee comment: "The abstract is rather short and could benefit from more results and interpretation so that the main message of the manuscript comes across right with the abstract."

*Response: Thank you, we have revised the abstract by expanding it to include key results and intepretation.*

2. "OSL Dating Please provide more detail on the OSL dating in the methods, especially how you have dealt with potential errors such as fading or incomplete bleaching. You should also provide reasoning (not only in the figure caption) why both OSL on quartz and IRSL on feldspar were measured but only OSL ages are discussed. One major novel statement of the manuscript is the presumably earlier recession of the ice from the study area based on OSL ages in comparison to earlier studies based on 14C radiocarbon ages. OSL ages usually come with higher uncertainties and standard errors than 14C-ages in this age range. However, the errors are not discussed here although they could easily move your OSL ages in the same age range as obtained by Bennike. Bennike's ages are right at the termination or after the termination of the Younger Dryas. In contrast, your ages are right within the Younger Dryas which is not prominent for widespread deglaciation but for the opposite. Please re-consider your arguments."

*Response: Thank you for your comment. We have revised the methodology and results sections concerning geochronology. OSL and IRSL ages are now presented with explanation. OSL ages are used in the geochronology while IRSL information is used to assess the bleaching of the deposits and the validity of the OSL ages. We have included an assessment of the error associated with OSL ages in the results section.*

3. "The authors use cryofacies terms that do not exist as such. (1) pore cryofacies, (2) layer cryofacies both are not cryofacies terms as it can be found or related to the literature. Pore ice is a certain ground ice type. Better use the term "structureless" as introduced in Murton and French 1994 and also used in French and Shur 2010. Then replace the new term throughout the manuscript. The same applies to "layer cryofacies" that should be replaced throughout by "layered". Be careful to also change this in the figure captions."

*Response: We agree that there are certainly inconsistencies in the literature concerning the terminology used to describe ground-ice structures. We have replaced 'layer' with 'layered' throughout the text. Several recent publications (Murton, 2013; Gilbert et al., 2016) use 'pore' in place of 'structureless' to describe frozen sediments with no visible ice. We selected 'pore cryofacies' for two reasons: (1) as we are combining sedimentology and cryostratigraphy we avoid confusion with the sedimentological meaning of structureless, and (2) we believe it to be a more intuitive term to describe sediments containing only pore ice. For these reasons we have elected not to change 'pore' to 'structureless'.*

**Specific comments**

*Thank you very much for the detailed comments, suggestions and corrections provided in the supplementary document. The in text format was very clear and we believe that we have incorporated all suggestions and corrections into the text. Two comments which we have not commented on in the "general comments" section are presented below.*

1. P2L12 "In general, are these ages calibrated or uncalibrated 14C ages BP? For radiocarbon ages, calibrated and uncalibrated ages in this age range make quite a large difference."

*Response: All referenced dates are calibrated and presented in calendar years BP. We use 'ka' to indicate kilo annum before present. If we were to present radiocarbon ages we would have used '$^{14}$C yr BP'. As the OSL ages are in calendar years we have used the calibrated ages for consistency. We are only using dates from a few publications and future researchers should be able to easily return to these publications and extract the radiocarbon ages if necessary.*

2. P5L17: "How was the volume of supernatant water determined? Please describe."

*Response: volumes of supernatant water and statured sediment were recorded from graduated beakers upon thawing – after Kokelj and Burn (2005). The methods section has been updated with this information.*

**References**

[revised manuscript text omitted]